# U.S. nursing home leadership experiences with COVID-19 and its impact on residents and staff: A qualitative analysis

**Catherine E. Dubé**[1]*, **Natalia Nielsen**[1], **Emily McPhillips**[1], **J. Lee Hargraves**[2,3], **Carol Cosenza**[3], **Bill Jesdale**[1], **Kate L. Lapane**[1]

1 Division of Epidemiology, Department of Population and Quantitative Health Sciences, University of Massachusetts Chan Medical School, Worcester, Massachusetts, United States of America, 2 Department of Family Medicine and Community Health University of Massachusetts Chan Medical School, Worcester, Massachusetts, United States of America, 3 Center for Survey Research, University of Massachusetts Boston, Boston, Massachusetts, United States of America

* Catherine.Dube@umassmed.edu

**Data Availability Statement:** Data collected for this research occurred prior to U.S. National Institutes of Health requirements for data sharing https://sharing.nih.gov/data-management-and-sharing-

## Abstract

### Objectives

To explore experiences of U.S. (United States) nursing home leadership during the COVID-19 pandemic in their efforts to address resident loneliness and social isolation and to elicit stories about personal and professional impacts on themselves and staff.

### Design

Qualitative inquiry via three optional open-ended questions appended to a national self-administered survey of American nursing home leaders was employed. Textual data was analyzed using an iterative reflexive thematic approach.

### Setting and participants

A stratified sample frame defined by facility size (beds: 30–99, 100+) and quality ratings (1, 2–4, 5) was employed. Web survey links and paper surveys were sent to 1,676 nursing home directors of nursing between February and May 2022.

### Results

Open text responses were collected from 271 nursing homes. Broad themes included: 1) Addressing needs of residents & families; 2) Challenges; and 3) Personal experiences of nursing home leadership/staff. Respondents described trauma to residents, staff, and leadership. Resident loneliness was addressed using existing and newer technologies and innovative indoor and outdoor activities. Residents experienced fear, illness, loss, and sometimes death. Isolation from family and lack of touch were particularly difficult. Regulations were seen as punitive while ignoring emotional needs of residents. Staffing challenges and pressures to do more with less created additional stress. Leadership and staff made significant sacrifices resulting in physical, social, and emotional consequences. Beneficial

policy/about-data-management-and-sharing-policies/data-management-and-sharing-policy-overview and elements of consent detailed in the preamble and body of the survey state specifically that participant answers will not be shared with anyone outside the research group. Ethics approvals did not include provisions for data sharing. Questions about access to If data for reanalysis or confirmation of results should be directed to the University of Massachusetts Chan Medical School ethics committee: Center for Clinical and Translational Science UMass Chan Institutional Review Board (IRB) 362 Plantation Street, Ambulatory Care Center, 7th Floor Worcester, MA 01605-0002 Phone: (508) 856-4261 IRB@umassmed.edu.

**Funding:** The work was supported by a grant to Dr. Lapane (KL) from the National Institutes of Health, National Institute on Aging (R01AG071692, 2021). https://www.nia.nih.gov/ The funders had no role in study design, data collection and analysis, decision to publish, or preparation of the manuscript.

**Competing interests:** The authors have declared that no competing interests exist.

outcomes included staff bonding, professional growth, and permanent implementation of new interventions.

## Conclusions and implications

New and creative interventions were successfully implemented to address social isolation and loneliness. Improved Wi-Fi and other nursing home infrastructure upgrades are needed to maintain them. Reimagining often conflicting overlapping federal, state, and local regulations, grounding them in good clinical judgement, and incentivizing performance improvement should be considered. Trauma experienced by staff needs to be addressed to deal with current and future workforce needs.

## Introduction

In the early stages of the COVID-19 (SARS-CoV-2) pandemic, a considerable proportion of COVID-19 deaths occurred in nursing homes around the world, with some facilities experiencing massive outbreaks [1]. Estimates of COVID-19 related deaths in long-term care facilities ranged from over 40% up to 80% of total deaths in some higher income countries [2]. In the US, nursing homes were among the first vectors for the spread of COVID-19 [3] and by August 2020, accounted for 41% of U.S. (United States) COVID-19 deaths [4].

Over 15,500 nursing homes serving approximately 1.3 million residents in the U.S. were affected [3, 5, 6]. Nursing homes were forced to isolate residents to slow the spread of COVID-19 thereby drastically worsening the loneliness and isolation already pervasive among this population [7–9].

In a previously published systematic review conducted by the authors [10] the scientific literature was methodically searched for evidence of the health impacts of loneliness among older adults living in nursing homes and other congregate living settings. Five databases were searched and articles were evaluated for inclusion. Out of an initial group of 370 full-text articles reviewed, 26 articles met inclusion criteria. A process of detailed review and synthesis led to the conclusion that most older adults living in congregate settings experienced at least some degree of loneliness with a few studies suggesting links to depression, suicidal ideation, and frailty. Social support appeared to moderate the association between loneliness and negative health outcomes and resilience and activities appeared to mediate this association [10]. This is consistent with the findings of the US National Academies of Sciences consensus study report (2020) which concluded that, although further research is needed, social isolation has been associated with premature mortality from all causes similar to other risk factors like sedentary lifestyle, obesity and smoking [11]. In some instances, prolonged social isolation may have led to death during COVID-19 restrictions when family visitation was curtailed [12]. The efforts and experiences of nursing home leaders attempting to address this problem during the COVID-19 pandemic was the focus of this study.

Despite the known damaging effects of loneliness among nursing home residents, the spread of COVID-19 prompted governments around the world to advise social isolation and nursing homes responded by severely reducing residents' social contact [13]. Recognizing the suffering this inflicted on their residents, nursing home leaders rapidly developed new solutions to deal with drastically worsening social isolation [14] and a range of interventions were quickly implemented. However, it was not until the pandemic was well under way (2021) that reports about interventions to address social isolation were published—a scoping review by

Bethell and colleagues [15], a rapid systematic review by Williams and colleagues [16], and a rapid review and synthesis by Ferdous and colleagues [17]. Consequently, for a lengthy period of time evidence-based solutions were not readily available to nursing home leaders and they had to rely on their wits and their personal and professional networks.

## COVID-19 and LTC staff

COVID-19 impacted nursing home staff worldwide in dramatic fashion [2]. In the United Kingdom, it was estimated that almost three quarters of working age COVID-19 deaths in England and Wales occurred among social care workers; and among social care workers, three quarters were workers in LTC (long-term care) or homecare [2, 18].

Studies around the world documented challenging COVID-19 working conditions in nursing homes and noted substantial psychological impact on the workforce [19–25].

Likewise, in the U.S., nursing homes faced unprecedented challenges. Expected to maintain routine health and social-emotional care throughout the pandemic, nursing home staff and leadership had to remain nimble, navigating rapidly changing state and federal COVID-19 guidelines whilst caring for their residents, all while bearing the emotional toll of working in stressful, potentially dangerous conditions [26]. The result was exacerbation of already problematic emotional overwhelm, burnout, and high staff turnover [25–27]. In fact, in the wake of the COVID-19 pandemic, U.S. nursing homes lost staff at levels exceeding any other health care sector, have had difficulty hiring new staff, and in some cases have been forced to limit admissions due to continuing staff shortages [28].

Despite the effects of the COVID-19 pandemic on the nursing home sector, only a few national studies [25, 29, 30] have documented how U.S. nursing homes pivoted to provide social-emotional care for their residents during these most challenging times, and how leadership, staff, and residents were impacted. This exploratory qualitative study attempts to further document these events in nursing home leaders' own words. Appended to a national survey, text-based stories were elicited from nursing home leadership about their COVID-19 pandemic experiences. Aims were to explore strategies employed to address resident loneliness and social isolation in U.S. nursing homes during the COVID-19 pandemic and to describe the personal and professional impact on nursing home leaders, their facilities, and their staff from the perspective of directors of nursing or administrators. Our goal was to report on these experiences from within the epicenter of an unprecedented crisis unique in modern times and to further collective understanding of these issues that may be pertinent to future pandemic preparedness.

## Materials and methods

### Human subjects

This study was approved as exempt research (45 CFR 46.104(d)(2) Educational tests/survey/ interview procedures, or observation of public behavior) by the University of Massachusetts Chan Medical School Institutional Review Board (Study 00000169) and the University of Massachusetts, Boston Institutional Review Board (Study 2021223). Research determined to be exempt is considered minimal risk and not subject to formal consent requirements (i.e., written documentation of consent). The survey was mailed or emailed to prospective participants, and elements of consent were included in the cover letter, beginning of the survey, and in the preamble to the open-ended questions used for this study. Importantly, it was noted that participation was voluntary, that participants could skip any questions they did not feel comfortable answering, and that answers are confidential. Further, it was stated that an individual's survey responses would not be shared with anyone other than the research team. The section

with open-ended qualitative questions was marked as optional and it was noted that these responses would not be linked to names but may be reported verbatim along with comments provided by other participants. Voluntary completion of the survey was considered evidence of consent. Participants are identified by participant number only.

### Research design

The phenomenon explored in this study is the response of nursing home leadership to the COVID-19 pandemic with a focus on interventions and innovations employed to address resident social isolation at a time of crisis. The conceptual model underlying the structure of the phenomenon under study is the 4-step COVID-19 crisis management model. The 4 steps in this model include *Calamity* or the onset of and initial reaction to the crisis (relies on preparedness) *Quick and Dirty* or initial planning and adapting (relies on agility), *Restart* or reorganizing to meet goals (relies on elasticity), and *Adapt to Next Normal* or adapting operations and developing new procedures for future flexibility (relies on redundancy) [31].

Given that study participants were directors of nursing or other nursing home administrators responsible for managing the operations and resident care at the Rapaccini 4-step crisis management model fits our initial conceptualization of the innovation process employed in response to the COVID-19 crisis and acute social connectedness needs of their residents at that time. The *Calamity* in this instance was the COVID-19 pandemic and its effects on resident care and well-being. This qualitative inquiry focuses on *Quick and Dirty* or immediate operational responses and adaptations, as well as fully implemented solutions or innovations (*Restart*) aimed at addressing pandemic-induced acute resident social isolation. It was anticipated that elicited stories would likely address both the *Calamity* and the solutions. It was also anticipated that some participants would address the final stage in this model (*Restart*) as they looked to the future. Finally, inquiry about the impact of these experiences on participants and their personal and professional lives was also planned.

This exploratory qualitative inquiry is guided by interpretivist epistemology [32] and was aimed at eliciting and better understanding participant experiences in their own words. Textual responses to optional open-ended survey questions were employed to enhance data collection from a diverse national (U.S.) sample and to maximize personal expression, allowing participants wide latitude to describe strategies, experiences, and the overall impact of the COVID-19 pandemic. Open-ended questions were appended to a national (U.S.) nursing home survey and were marked "optional." Open responses and confidentiality protections were intended to reduce response bias. This approach was selected for efficiency and expediency at a time when the COVID-19 pandemic was beginning to wind down and pandemic experiences were still fresh. An iterative reflexive thematic approach to analysis was employed.

The quantitative survey (described elsewhere [33]) that preceded qualitative open-ended questions was a self-administered questionnaire completed by directors of nursing or administrative leaders who were working at U.S. nursing homes during the COVID-19 pandemic. These leaders assessed the experiences of staff and residents during the pandemic both before and after vaccines were widely available.

### Participant selection and survey process

Explained elsewhere in detail [33], participant selection and survey process are briefly described here. In preparation for survey administration, a purposive sample was constructed. First, a list of all nursing homes rated by the U.S. Centers for Medicare & Medicaid Services (CMS) as of August 2021 was acquired. Small nursing homes with <30 beds were excluded because they differ significantly from larger nursing homes both in terms of operations and

resources. (Average size of a U.S. nursing home in 2022 = 106 beds [34]). A total of 14,613 nursing homes were stratified by size and CMS quality rating into six strata (30–99 or 100 + beds; 1, 2–4, or 5-star ratings) with 283 nursing homes selected in each stratum. Since the probabilities of selection varied among strata, survey design weights were used for the quantitative analyses to provide nationally representative estimates.

Data were collected between February and May 2022 by the Center for Survey Research at the University of Massachusetts Boston. Directors of nursing for each nursing home were identified and contact information was collected via internet searches and phone calls placed directly to selected nursing homes. Links to the online survey were sent to those with email addresses. Those who completed the survey from the online link were sent $45. Paper surveys including a $5 incentive were mailed to all email non-responders and those without email addresses. Those who completed the paper survey received an additional $40 compensation.

Ultimately, the sample used for the qualitative analysis (n = 271 from a total of 504 survey responders) was a convenience sample of voluntary participants derived from the purposive/ nationally representative sample assembled for the national nursing home survey. Reasons for nonresponse to open-ended questions are unknown. No participants withdrew.

## Survey content

Three optional open-ended questions were included aimed at eliciting respondents' thoughts, views, and opinions about pandemic experiences in their own words. Open-ended questions were drafted and pretested (described below) and reviewed and finalized by the research group. Final questions were:

**Preamble**: Please reflect upon how you, your facility and direct care staff responded to resident loneliness and social isolation during the pandemic. We want to hear your candid thoughts and opinion. Your responses will not be linked to your name but may be reported verbatim along with comments provided by other participants.

1. Please describe in detail one successful strategy your nursing home used to help residents reduce loneliness and social isolation during COVID-19. Please tell us the story of this strategy and how it worked in your nursing home. How did you know it worked? What challenges did you overcome? Tell us how you overcame these challenges.

2. Was there anything that could have been done differently to help residents reduce loneliness and social isolation at your nursing home during COVID-19? If you had to do it all over again, what would you do differently?

3. How did your professional experiences with the COVID-19 pandemic affect you? What event(s) in your workplace deeply affected you? Please describe the event(s) and the impact on you personally and professionally.

The quantitative survey included 79 questions about staff and residents. Eight demographic questions about the responder were also included and were used to characterize the population of qualitative responders. Topics addressed by the quantitative survey included stress and concerns about direct care staff burnout, intentions to leave, and staffing challenges; social isolation among residents; and what nursing homes did to address resident social isolation. Questions primarily employed rating scales or multiple-choice response sets [33].

The survey and open-ended questions were pretested using cognitive interviews [35]. Four nursing home directors of nursing (DONs) participated in testing. Each DON was asked to complete the survey prior to a Zoom video call where they discussed their survey

experiences. Revisions primarily focused on quantitative questions and response options. Qualitative questions performed as intended and no revisions to qualitative questions were suggested.

### Data collection and setting

Open-ended questions included at the end of the online survey allowed for free text responses up to 20,000 characters and the mailed survey provided two inches of space for hand-written responses for each question. A research assistant/consultant blinded to participant identifiers transcribed text responses from paper surveys into a digital format. Transcripts were reviewed for accuracy.

Surveys were sent to work emails or mailed to work addresses. However, participants could respond to the questions anywhere and it is unknown who might have been present when participants wrote their responses.

### Qualitative analytic approach and rigor

CD (a professor experienced in qualitative research who teaches a graduate course in qualitative research methods) led the qualitative analysis process. CD, EM, and NN conducted the analysis. A modified inductive thematic analysis approach, guided by principles of experiential reflexive thematic analysis [36] was employed. An iterative process of emersion/crystallization [37] was used by individual analysts for initial presentations of interpretations to the research group prior to preparation of a consensus summary. These approaches were selected to reduce bias while determining the meaning, quality, preponderance, and range of expressions. For each open question, summaries and lists of themes were independently constructed followed by a consensus summary. Each consensus summary was then assessed to determine if detailed thematic coding was required. Two themes, RESIDENTS and BENEFITS were identified as both cross-cutting and relatively rare. For these themes, NVivo qualitative analysis software was used [38] and all responses were examined and coded as appropriate. Consensus summaries were also prepared for these 2 cross-cutting themes. Overall, three broad themes corresponding to the three open-ended questions were identified:

1. Addressing the needs of residents and families–reducing social isolation and increasing connection

2. Challenges

3. Personal experiences of nursing home leadership and staff

Results below were derived from independent summaries, consensus summaries, and code reports and summaries. Draft results were reviewed by the analysis group and the research team and the final version was agreed upon.

(NOTE: For additional information about research procedures please see S1 Appendix 1. Research Protocol.)

## Results

### Saturation

CD, EM and NN completed initial reviews of the complete set of responses for each open-ended question. The consensus opinion was that saturation was reached midway through each response set.

## Participant characteristics

From an eligible sample of 1,676 U.S. nursing homes, 504 surveys were collected (response rate 30%) of which 271 (54% of the 504 respondents) answered at least one open-ended question. Respondents were mostly women, White non-Hispanic, and Directors of Nursing. Those who responded to open-ended questions tended to be older, have a longer tenure at their current nursing home, and have more time in a leadership position than non-responders (see Table 1). Respondents represented 44 U.S. states (Table 2) and each respondent represented a different nursing home. The number and length of open-ended responses are summarized in Table 3. A summary of qualitative results is presented in Table 4 and illustrative quotes for each theme and subtheme are provided in Table 5.

## Theme 1: Addressing the needs of residents and families–reducing social isolation and increasing connection

**Residents–profound effects of the pandemic.** Although not directly questioned about resident experiences, survey respondents discussed residents within the context of their own professional and personal reactions and actions taken to address resident needs. They noted that nursing home residents struggled with both direct effects of COVID-19 and associated changes to their routine and environment. Outbreaks in facilities often resulted in fear, suffering, and death. Several respondents described their residents as family–and other residents and staff became surrogate family during visitation restrictions. Social isolation due to COVID-19 and prolonged lack of human touch were seen as particularly harmful to residents, perhaps leading to depression. Residents who suffered loss of neighbors and friends sometimes experienced survivor's guilt. Some respondents described how residents' basic needs were not being met during the pandemic. One participant commented: "*Residents were forgotten, not showered, rooms were dirty" (Participant 5004)*. Other respondents mentioned a rise in: 1) isolation, loneliness, and fear of loneliness; 2) behavioral issues (trouble eating or sleeping, aggressive behaviors, crying, etc.); 3) communication difficulties with masks/face shields; 4) resident confusion due to room changes for isolation; and 5) for those residents who didn't understand, fear of staff dressed in full Personal Protection Equipment (PPE). Some participants raised concerns about residents dying of broken hearts: "*I feel like we lost more to broken hearts than we did before COVID" (Participant 6122)* and for the potential long-term impact on residents' mental health: *"I believe we will see mental health fall out from those first several months of covid for many years to come" (Participant 4089)*. Residents in facilities with adequate staff and minor or no COVID outbreaks had less severe consequences.

**1.2 Family–resident families' experience & importance of family connection.** Residents' basic human need to be connected with family was described by respondents in profound ways: "*So many of our residents voiced they would rather die than not get to spend time with their family" (Participant 2043); "Many [dementia patients] cried because they thought their family had died or no longer loved them. It was heartbreaking to watch many residents give up and die from a broken heart" (Participant 6049)*. Facilitating this connection using a variety of methods was seen as key. Many respondents indicated that residents were notably happier and healthier (e.g., eating more) when they were socially engaged and able to communicate with family. Separation from family was described as heartbreaking for staff and detrimental to residents especially for those with dementia who did not know why their family was not visiting. Some characterized this separation from family as more lethal than COVID-19. Interventions aimed at increasing family connection were seen as successful.

**Technology solutions.** Respondents described addressing social isolation and loneliness during COVID-19 on multiple fronts using both existing technology and technology new to

**Table 1. Characteristics of participants–Responders to open-ended questions and non-responders.**

| | Responded ≥1 open-ended questions n = 271 (53.8%) | Did <u>not</u> respond to any open-ended questions n = 233 (46.2%) |
|---|---|---|
| | *Percentage* | |
| **Age (years)** | | |
| 18–40 | 18.9 | 22.8 |
| 41–50 | 33.2 | 31.8 |
| 51–60 | 30.3 | 24.0 |
| >61 | 15.5 | 10.7 |
| Not Ascertained | 2.2 | 2.6 |
| **Gender** | | |
| Woman | 91.1 | 88.0 |
| Man | 8.9 | 9.9 |
| Something else or Not Ascertained | 0 | 2.29 |
| **Race/Ethnicity** | | |
| Hispanic | 3.3 | 4.7 |
| White, Non-Hispanic | 80.4 | 78.1 |
| Black, Non-Hispanic | 9.6 | 9.0 |
| Asian/Pacific Islander | 5.2 | 3.9 |
| Other/Mixed Race | 1.5 | 2.2 |
| Not Ascertained | 0 | 2.2 |
| **Length of Employment at Current Nursing Home** | | |
| <1 year to 4 years | 40.2 | 44.7 |
| 5 to 9 years | 19.6 | 19.7 |
| 10 to 14 years | 13.3 | 15.0 |
| 15 or more years | 26.9 | 19.3 |
| Not Ascertained | 0 | 1.3 |
| **Job Title** | | |
| Director of Nursing | 85.6 | 83.7 |
| Nursing Home Administrator | 10.3 | 9.4 |
| Other | 4.0* | 5.6** |
| Not Ascertained | 0 | 1.3 |
| **Length of Years in Leadership Position** | | |
| <1 year to 4 years | 18.8 | 30.5 |
| 5 to 9 years | 27.3 | 25.8 |
| 10 to 14 years | 17.3 | 17.2 |
| 15 or more years | 36.5 | 25.8 |
| Not Ascertained | 0 | 0.9 |

*Other job titles include: Infection Preventionist, Nursing Home Administrator, MDS Coordinator, Charge Nurse, Health Services Coordinator, Licensed Practical Nurse, Interim Director of Nursing, Nurse Administrator, Regional Chief Nurse Executive and Staff Development/Infection Control Nurse Manager.

**Other job titles include: Chief Nursing Officer, Infection Control Preventionist, Staff Development Nurse, Resident Care Manager, Social Services, Vice President of Health Services, Vice President of Clinical Operations, Business Office Manager, Human Resources, and Case Manager.

the nursing home setting. The use of newer technology (such as iPads, FaceTime, etc.) to help residents communicate with their loved ones was universally regarded as successful. Some respondents noted the need for additional funding and staff to implement and maintain use of

**Table 2. Number of respondents by U.S. state (n = 271).**

| State | n | State | n | State | n | State | n | State | n | State | n |
|---|---|---|---|---|---|---|---|---|---|---|---|
| AK | 0 | FL | 12 | LA | 5 | NC | 3 | OK | 9 | VA | 6 |
| AL | 2 | GA | 3 | MA | 5 | ND | 3 | OR | 1 | VT | 0 |
| AR | 4 | HI | 1 | MD | 2 | NE | 5 | PA | 15 | WA | 0 |
| AZ | 2 | IA | 13 | ME | 6 | NH | 6 | RI | 0 | WI | 4 |
| CA | 12 | ID | 1 | MI | 12 | NJ | 7 | SC | 1 | WV | 0 |
| CO | 9 | IL | 10 | MN | 22 | NM | 0 | SD | 3 | WY | 2 |
| CT | 6 | IN | 5 | MO | 9 | NV | 1 | TN | 3 | Un-known | 1 |
| DC | 0 | KS | 13 | MS | 2 | NY | 9 | TX | 13 | | |
| DE | 3 | KY | 4 | MT | 3 | OH | 9 | UT | 4 | | |

these technologies. It was noted that some residents had challenges using technology due to cognitive, vision, and/or hearing impairments. Existing technology used in new ways included internal closed-circuit TV, smart TVs, overhead public address systems, and telephones. One respondent mentioned the successful use of robotic pets *(Participant 1206)*.

**Distanced indoor activities.** Respondents described indoor group activities in hallways and in rooms large enough to accommodate social distancing. Hallway activities most frequently mentioned included hallway Bingo as well as sing-alongs, ice cream socials, and hallway parties. Several mentioned the importance of prioritizing the reinstatement of communal dining as soon as it was safe and feasible. Other socially distanced indoor interventions included pet therapy, music therapy, art therapy, compassionate care visits, staff-resident interactions, and bereavement services.

**Staff connection.** Respondents described several different initiatives aimed at increasing social contact between staff and residents. Some hired new staff for this purpose, while other facilities scheduled existing staff regardless of role to socialize with residents. Others assigned members of leadership to this task. Some in leadership took on other direct care roles where staffing was short. One-on-one quality social time with staff was described as *"filling the gap"* *(Participant 1043)* in the lives of residents, intended to compensate for loss of in-person family visit time and to augment efforts to connect residents with families through other means. Staff shortages made these interventions difficult although it was noted that some staff worked extra hours or donated their time to provide social time with residents. A few respondents mentioned the importance of the use of clear masks to enhance communication, allowing residents to see staff's facial expressions.

**Outdoor activities.** Outdoor activities mentioned included window visits, outdoor family visits, music on the patio, van rides, and family or community participation in drive-bys and parades. Window visits were mentioned most often although some facilities without ground floor rooms were unable to use this strategy. Significant projects to create outdoor spaces for distanced activities were also mentioned. Several suggested that nursing homes did not have sufficient resources for such projects. A few mentioned charitable donations from the community.

**Table 3. Number and length of open-ended responses.**

| Open-Ended Question | N | Response Length (Words) | | |
|---|---|---|---|---|
| | | Range | Mean | SD |
| **1. Successful strategies to address loneliness/social isolation** | **240** | **3–156** | **43.6** | **31.5** |
| 2. What would you do differently? | 229 | 1–125 | 30.0 | 22.8 |
| 3. Effects on you personally/professionally | 264 | 1–488 | 70.4 | 68.1 |

**Table 4. Results summary table.**

| Thematic Area | Noted in Participant Responses |
|---|---|
| **1. Addressing needs of residents & families–reducing social isolation and increasing connection** | |
| 1.1 Residents–pandemic effects | • Residents endured outbreaks, illness, death<br>• Pandemic fear, loss, survivor's guilt<br>• Lack of human touch<br>• Resident social isolation<br>• Isolation leading to resident behavioral issues, depression<br>• Communication difficulties due to masks, face shields, etc.<br>• Pandemic stresses led to erosion of quality of care |
| 1.2 Interference with family connection due to pandemic restrictions | • Dementia patients feeling abandoned<br>• Resident loss of appetite<br>• Potentially lethal for some residents<br>• Stress and "heartbreak" for staff |
| 1.3 Technology solutions to address social isolation | • iPads, FaceTime<br>• Existing closed-circuit TV, smart TVs<br>• Overhead public address systems<br>• Telephones |
| 1.4 Distanced indoor activities | • Hallway activities and hallway parties<br>• Reinstating communal dining<br>• Therapies–pet therapy, music therapy, art therapy<br>• Bereavement services, compassionate care visits |
| 1.5 Staff connection | • Scheduling social contact between leadership/staff and residents<br>• Compensating for loss of family visits<br>• Some staff volunteered/donated their time<br>• Challenge: Staff shortages |
| 1.6 Outdoor activities | • Window visits, outdoor visits<br>• Patio activities, van rides<br>• Community drive-bys<br>• Challenge: Lack of funding |
| 1.7 Wishlist–what they wished they had or had sooner | • Nothing–some wouldn't change what they had done<br>• Allowing family visits sooner with personal protective equipment<br>• More distanced activities<br>• Facility improvement and improved or more technology |
| **2. Challenges** | |
| 2.1 Frustration with government regulations | • Regulations were rapidly changing "moving target"<br>• Lack of appreciation for resident mental health<br>• Resulted in detrimental resident social isolation<br>• "Impossible expectations" from oversight agencies<br>• Fines or citations for deficiencies was punitive |
| 2.2 Staffing challenges | • Difficult climate: fear of COVID-19, long hours, stifling regulations, and heavier workload<br>• Increased resignations<br>• Staff shortages limited programmatic innovations |
| 2.3 Pressures–Internal and external pressures | • Negative portrayal in media<br>• Rapidly changing health emergency<br>• Family upset and demands<br>• Personal and staff stress and burnout<br>• Additional reporting requirements |
| **3. Personal experiences of nursing home leadership and staff–effects of the pandemic** | |
| 3.1 Physical, social, and mental impacts and consequences | • Stress, overwhelm<br>• Hopelessness, depression, anxiety, loneliness<br>• Loss of sleep, weight gain<br>• Isolation from family, marital/relationship discord<br>• Missed family occasions and milestones<br>Lost job satisfaction |

(*Continued*)

**Table 4.** (Continued)

| Thematic Area | Noted in Participant Responses |
|---|---|
| 3.2 Beneficial, unanticipated positive effects | • Made them stronger<br>• Bonding with colleagues and staff<br>• Developed new leadership skills<br>• New appreciation for activities and psychosocial wellbeing of residents |

**Wishlist–what they wished they had or had sooner.** Many respondents felt as though, if they were to go back in time, there was really nothing they could have done differently. Others wished that they had implemented certain things sooner and/or more often, such as allowing family visits while wearing PPE and arranging more socially distanced activities for residents. Another common thread among respondents was the need for facility improvements including increased availability of technology and a strengthened Wi-Fi connection to improve virtual communication between residents and their families.

## Theme 2: Challenges

**2.1 Frustration with government regulations.** Respondents felt limited by rapidly changing COVID-19 regulations that were characterized as *"ever changing" (Participant 5139).* Many felt that regulations seemed to ignore the mental health needs of nursing home residents by creating social isolation that was seen as very detrimental. Keeping up with regulations was described as confusing and time-consuming, requiring time and attention that would otherwise be used for resident care. Some respondents complained of impossible expectations from oversight agencies who were seen as out of touch with reality. Others felt victimized by these agencies as they faced harsh attitudes or punishment in the form of fines or citations if deficiencies were identified. A few respondents expressed strong disagreement with COVID-19 related protocols and distrust of government authority.

**2.2 Staffing challenges.** Staffing was frequently identified as a serious concern that far exceeded any previous staffing challenges. Respondents described several pandemic-related factors that made staffing difficult, including fear of COVID-19, long hours, stifling COVID-19 rules and regulations, and extra duties resulting in a heavier workload and greater demands. Staff shortages and hiring challenges due to stress, PTSD, staff burnout, turnover, and short staffing due to staff illness and quarantine requirements were also mentioned. One respondent reported that vaccine mandates also led to staff resignations: *"The vaccine mandate was very difficult with the loss of full-time personnel" (Participant 2191).* Resulting staff shortages limited what could be accomplished during COVID-19 at a time when residents required additional care and support.

**2.3 Pressures–internal and external pressures.** Respondents were distressed by the negative portrayal of nursing homes in the media and reported feeling blamed by the media and society for the crisis in nursing homes at the time. A few respondents felt they *"went from heroes to murderers" (Participant 2191)* in the media. *"We have very intelligent, caring individuals working in homes but the media made us look horrible—felt sadness over this, anger but also motivated to continue to provide the best care for our residents as possible!" (Participant 2032)—* and some reported that the media had been hovering around their facilities. Other key stressors included adjusting to a rapidly changing emergency while having to scramble to respond to changing regulations and guidelines, maintaining adequate staffing, dealing with families' upset and demands during COVID-19 visitation restrictions, and dealing with staff stress and burnout:

**Table 5. Themes, subthemes and illustrative quotes derived from open-ended questions included on a COVID-19 survey of nursing home administrators/directors of nursing.**

| Themes | Subthemes | Illustrative Quotes |
|---|---|---|
| 1 Addressing the needs of residents and families–reducing social isolation and increasing connection | 1.1 Residents–profound effects of pandemic | *To see the residents so sad over the isolation fear and loneliness then have to deal with unexpected death. . . unexpected death in the otherwise healthy individuals was just devastating. (Participant 4030)*<br>*Challenges were calming the fear to know it was ok to be in the doorway for activities. Residents losing their neighbors and friends to COVID. It was survivor guilt for residents. (Participant 2012)*<br>*Watching residents become depressed from not being able to hold, touch, or hug their family members. Watching residents become frustrated because they couldn't understand what was being said to them because of mask/shields. Residents who were used to going out with family not understanding why they couldn't. Confusion. . . as they relearn where their rooms were from having to move people around to isolate. (Participant 6180)*<br>*Somedays we struggled (and still do on some days) to get the very basic of care done. (Participant 4099)*<br>*The very worst part of COVID was isolation. And really the isolation was not the cure-all as hoped by those guiding us. The isolation really killed more residents than COVID in our building. (Participant 2012)* |
|  | 1.2 Family–resident families' experience & importance of family connection | *The isolation from family was extremely detrimental to our residents and I feel led to the poor outcomes of recovery in those who passed with COVID time we had our first outbreak most of our residents were already suffering from the isolation from family. So many of our residents voiced they would rather die than not get to spend time with their family. (Participant 2043)*<br>*I dealt with dementia residents that didn't understand why their family was no longer coming to visit them. Many cried because they thought their family had died or no longer loved them. It was heartbreaking to watch many residents give up and die from a broken heart. (Participant 6049)* |
|  | 1.3 Technology solutions | *Our facility looked toward Televisit as an ideal strategy to keep patients in touched with their families/loved ones. It took significant coordination between Information Technology, Social Workers, Hospital Security and Nursing. Some of the challenges were scheduling and staffing availability for the Televisit. The staff involved got more familiar with setting up the new process and got more familiar with the technology. It worked well with patients who had family members. It alleviates some of the concerns and questions from family members. (Participant 1269)*<br>*Use of technology was very successful. . . We utilized our internal TV channel as well to broadcast worship, campus updates, exercise programs, etc. across our life plan community, including the nursing home. (Participant 5008)* |
|  | 1.4 Distanced indoor activities | *The activity department started going around twice a week or so and playing music on a cart in the hallway and giving out ice cream sundaes. The residents would sit in the doorways and watch the staff dish up the ice cream, listen to the music and chit chat with staff as they were preparing it. I think the residents really loved this and gave them something to look forward to. (Participant 5059)*<br>*As soon as possible, we resumed dining as a group. To accomplish this with social distancing we seated residents at least 6 feet apart and expanded serving times to accommodate residents' participation in communal dining. We know it worked because resident complaints went down slightly and their intakes seemed to improve. Our biggest challenge with implementing this change was kitchen staffing; often did not have a dietary aide. We overcame that challenge by having other staff (office, nursing, etc) deliver meals to residents at their tables. (Participant 5110)* |
|  | 1.5 Staff connections | *As a team we immediately recognized the potential isolation; therefore, our team made a conscious effort to fill the gap ourselves in the day-to-day life of our residents thinking of them as our own family. We foster connection with relatives thru face time as many times a day as necessary or requested, engaging 3–4 individuals going around each resident offering the connection with their loved ones or spending a one-on-one quality/social time with them. (Participant 1043)*<br>*We provided 1:1 visits on a schedule for residents and assigned certain staff members to visit with residents. This helped them feel like people were coming in to talk to them, not ONLY to provide care. The biggest challenge to this was that we were already so short-staffed, and finding staff do this was difficult. We had to rearrange the schedule sometimes as a result. (Participant 4142)* |
|  | 1.6 Outdoor activities | *We tried to make window visits as pleasant as possible for our residents and family in the early days of the pandemic. We placed room numbers on the outside windows throughout the facility to assist in located loved ones. We placed our residents with COVID in the window beds versus the door beds, this way they could visit from their beds if very ill. Staff assisted residents with cell phones to help with communication as the windows were expected to stay closed. (Participant 1056)*<br>*We live in a small rural town-we posted on the city page for families with children (many were doing virtual learning at that point) to do "drive-by's" on certain days/hours-we opened resident curtains and the children waved/sang to the residents on these days-residents were so excited and learned quickly the "days/times" the children would come! The adults joined in as well-the entire community benefited! (Participant 5279)* |
|  | 1.7 Wishlist–what they wished they had or had sooner | *We did the best we could with what we had and abiding by the regulations set in place. We could have done more hallway activities. In the beginning when there was a lot of unknown and it was hard because everyone had to be socially distanced. It would have been nice to do more small group activities but staffing was limited. (Participant 1108)*<br>*Looking back, I wish we could invest in more technology to help with family visits. Maybe something like an Amazon Echo Show so families could just drop in and talk to residents whenever and for however long they wanted, instead of having to schedule a limited-time videoconference with a staff member. I think residents and families would have felt more in control. (Participant 3175)* |

*(Continued)*

**Table 5.** (Continued)

| Themes | Subthemes | Illustrative Quotes |
|---|---|---|
| 2 Challenges | 2.1 Frustration with government regulations | *The constant change in rules and strategies and the enormous resources needed to comply with them was the most difficult thing to deal with. It took the most important resource we have for our residents, our time, from them. Prior to COVID I found time to spend with our residents daily. Since the pandemic 50–75% of my time is spent on issues/ regulations surrounding COVID each week.* (Participant 1174)<br>*A seed of resentment to oversight was created. The ability for non-clinical authorities to have credibility is broken. Labeling front line staff as "heroes" or "essential" has also backfired. Staff now feel used in a game of liability and blame. Mandates have further destroyed any goodwill between clinical staff and administration. It seems as if the entire industry was torpedoed and then sacrificed for the benefit of established government oversight.* (Participant 4044)<br>*LTC workers were expected to be the sole support to our residents, all while being placed under significant and at times unachievable restrictions that prevented us from being the support that our residents deserved.* (Participant 1020) |
| | 2.2 Staffing challenges | *Staffing the hospital in the start of the pandemic was deeply troubling. The level of anxiety and uncertainty had a toll on everyone. There were staff quitting, staff who were sick and staff who were truly afraid. I was concerned for my nursing managers and staffers who struggled especially after hours to staff the hospital. The phone calls were ongoing after hours from challenges faced during the emergency. Working long hours and following up with staff notifications was hectic.* (Participant 1269)<br>*The stress of the workload, staffing needs, resident isolation effects, family complaints, and much more were overwhelming to say the least. Nursing and healthcare have dramatically changed since the pandemic began and although we seem to have grown accustomed to the high stress positions, it is making it much more difficult to retain staff in this field and keep from staff burnout, including my own.* (Participant 6074) |
| | 2.3 Pressures -internal and external pressures | *We also continue to be blasted in the media as the "bad guys" all the time even though the requirements placed on us are much more strict than any other area of healthcare even though our residents come from and go to the hospitals and clinic settings. We stopped following the science of this a long time ago and are now just following the money and creating a way for even more tight guidance to be placed on LTC so when the survey teams come in they can issue citations and hand out fines that allow them to recoup money we have already been waiting 2 years to get from our cost report submissions. We continue to be asked to do more with less and no one entered LTC for the money. We came because of our love for the seniors we serve but, sadly, many are leaving because the pressures in this area of senior care are simply not worth the personal toll it can take on us.* (Participant 4156)<br>*[The] overwhelming frustration at sending residents to our local hospital and being told, basically, that despite the fact that these residents were full code that the hospital treating them would be a "waste of resources" and then having them sent back to the facility without availability of hospice or medications for comfort care. Several of my nurses were berated by the physicians at the local hospital for even sending these residents to the ER causing them to question their judgment in seeking emergency care for them.* (Participant 5110) |
| 3 Personal experiences of nursing home leadership and staff | 3.1 Physical social and mental impacts and consequences | *I personally had an emotional breakdown. The negative focus I had on numbers, the regulations that restricted everything and the burnout caused by staffing shortages caused me to seek out help.* (Participant 1006)<br>*When I recovered in June 2020 and was rushing back to go help my staff at work, it almost cost me my marriage. My wife felt I was more interested in being a hero than protecting my family. Took time to convince her.* (Participant 2256)<br>*Our facility was a battle zone and staff were working so hard to provide care to our residents. Staff were out with COVID, it was heartbreaking and exhausting. Emotionally it was very hard to watch residents pass and see how this impact staff members. . . I have never in my 45 years of employment had to endure anything so traumatic.* (Participant 6104)<br>*I really would rather be a truck driver at this point. I have lost my love for nursing.* (Participant 1153) |
| | 3.2 Beneficial, unanticipated positive effects | *Staff: from all departments, all pay scales and scopes worked together to cross-over scopes to meet the needs of residents. Our salary staff were working their positions and then coming in and working the floor on NOCs, weekends, etc. to support frontline staff and serve our residents. Families nor the media saw that. The true grit, dedication, and passion it took to "make it through" this pandemic.* (Participant 1176)<br>*I still have a list I keep of positive outcomes from this experience as motivation (intentional time with family, efficiencies of virtual meetings, better understanding of front-line staff's work, their appreciation for administration, deeper mindfulness for integrity and accountability).* (Participant 1172)<br>*[I] have also been able to first-handedly witness teamwork, compassion for the elderly, resiliency, dedication and strength.* (Participant 1176)<br>*Our facility and team are visibly happier as we come out of the pandemic. Mainly because we are starting to be able to get back to the reasons we are in the positions we are to begin with- the bonds, the fun, the development of staff, the group/ community activities, the team, which includes everyone, our residents, staff, and families!* (Participant 3065) |

*"Most stressful was developing policies and plans to mitigate and protect residents immediately after COVID was discovered in America. Next stressful was staying abreast of the ever-changing CDC guidelines. Next was finding staff to do the extra duties entailed in testing staff and residents up to twice per week for weeks on end then counting and reporting this information accurately every week to [the National Healthcare Safety Network]. Lastly, carrying the burden of trying to safely staff the facility. . . when EVERYONE is burnt out. . . and no one wants to endure it anymore" (Participant 1278).*

Some respondents described the challenge of day-to-day survival, frustration with local hospitals, and pressure to do more with less as new protocols and interventions were needed and staffing was short.

## Theme 3: Personal experiences of nursing home leadership and staff–effects of the pandemic

**3.1 Physical, social, and mental impacts and consequences.** The first two years of the COVID-19 pandemic were described as extremely stressful, affecting the mental and physical health of overworked staff. The majority of respondents felt they were affected both professionally and personally by the pandemic experience, and it took a toll: *"The pandemic was the most stressful situation I have encountered in [>20] years of nursing"* (Participant 1278); *"Everyday was a challenge and I felt hopeless"* (Participant 2028); *"I personally had an emotional breakdown"* (Participant 1006). Respondents reported physical, emotional, and social consequences including feelings of stress and overwhelm, depression, anxiety, loss of sleep, weight gain, and loneliness. Some indicated they required antidepressants, anxiolytics, or other pharmacotherapies. Isolation from family and tension with spouses/marital discord provided added stress. Stretched thin at their jobs, taking on additional responsibilities, working long hours, and unable to take a break, respondents often missed holidays, vacations, and important family milestones. Many respondents talked about leaving nursing/administration, advising others not to enter the field, or not going into long-term care in the first place if they could go back in time: *"I have lost my love for nursing. I remain only for the residents and my colleagues"* (Participant 1153).

**3.2 Beneficial, unanticipated positive effects.** A few respondents described "silver linings" to the COVID-19 experience. One felt the experience made them stronger, and another described bonding among staff from pulling together as a team even when short-staffed and overworked. The result was a strong emotional connection among the staff. Other respondents felt the COVID-19 experience led to professional growth: *"I grew as a professional from dealing with extremely difficult times, including, but not limited to, communication skills (dealing with difficult and not understanding residents/ families/staff), leadership skills, infection control skills, improving ever changing processes and policies"* (Participant 3065). COVID-19 seems to have made combatting resident loneliness a top priority in a way it had not been before. One respondent described how it became clear during COVID-19 that the Activities Department was incredibly vital and appropriate as quality activities were key to the psychosocial wellbeing of residents. Plans to continue new and innovative activities and interventions aimed at resident wellbeing were described by several respondents.

## Discussion

The COVID-19 pandemic severely challenged both nursing home staff and residents. In response to immediate problems, numerous strategies were implemented that also have potential to improve care of nursing home residents in the future [39]. Our study aim was to

document the experiences of nursing home leaders and the strategies they employed to address a unique crisis in nursing home care while supporting their staff and residents' health and social needs. U.S. nursing home directors of nursing and administrators in our national sample described a wide range of creative tactics employed to address the needs of residents despite severe resource and staff limitations. Rapid adaptation to a constantly changing emergency situation relied on impressive levels of dedication and resilience. Some respondents indicated that the innovations they implemented over the course of the pandemic would be continued, and that the pandemic experience, although extraordinarily stressful, led them to improve their leadership, infection control, and management skills.

## Conceptual model

Overall, responses from participants generally reflected the Rapaccini 4-step crisis management model [31]. Responses revealed an acute understanding of the problem (*Calamity*), agility in formulating a rapid response (*Quick and Dirty*), and adaptation (*Elasticity*). Some participants volunteered plans for a future where innovations would be continued (*Adapt to Next Normal*). However, unplanned government mandated changes, often accompanied by frustrations or confusion, were a more difficult fit. These imposed changes to nursing home operations often had unclear clinical rationale and had to be rapidly implemented oftentimes without adequate planning, communication, or evaluation of resources. Further, the rapidly changing nature of nursing home conditions during COVID-19 and cascading crises in some facilities appeared to exceed the bounds of the Rapaccini model. Long-standing deficiencies in U.S. nursing homes in areas such as patient care and safety, infection control, staffing, and oversight [40] exacerbated difficulties in adapting to a new normal.

## Addressing needs of residents & families

During COVID-19 pandemic restrictions, nursing home residents were described as fearful and isolated and, for some, this led to behavioral issues, depression, communication challenges and ultimately the erosion of quality of care. Without visitation and subsequent loss of family connection, some residents (particularly those with dementia) felt abandoned, stopped eating, and some died. In the present study, many interventions to address resident social isolation were adaptations of pre-pandemic techniques but conducted either outdoors, inside with social distancing, or by using technology. Video calls were often mentioned as a way to address patient isolation and separation from family. However, the efficacy of video calls in alleviating loneliness has been challenged [41]. Several strategies from published literature with perhaps the most evidence supporting their use were not mentioned by respondents in the present study. These included pain management and addressing hearing and vision loss [15], mindfulness, meditation, laughter therapy, art discussions, Wii computer gaming [16], physical exercise, changes to the nursing home environment to promote socialization, therapeutic touch, and regular screening for isolation [17]. Other strategies that were mentioned by respondents have been documented in other studies or in online trade communication and include window visits, use of phone calls and other familiar technologies, and socially distanced indoor and outdoor activities [17, 30, 42, 43]. Difficulties using newer technologies that may be unfamiliar to residents have been noted including insufficient equipment and infrastructure, implementation challenges, difficulty adapting technology to meet resident needs [44], and a need for staff assistance in its use [45].

## Challenges

Although there was variability in the present study among participant responses, for many the scope of the emergency appeared to be extremely challenging and, in some cases, severely

emotionally taxing. This is largely consistent with other research. Described as "complex and stressful," challenges identified by nursing home staff included re-use of personal protective equipment (PPE) and its associated risks, keeping up with changing and at times contradictory government guidance, and dealing with staffing shortages [25]. In the present study, PPE issues were not raised perhaps due to the timing of data collection in 2022 when the PPE shortage had been addressed.

Nursing homes in the U.S. have historically suffered from staffing challenges due to difficult working conditions, negative views of nursing homes, low compensation, and poor benefits [40]. Nursing home work in the U.S. is a dangerous profession with among the highest death rates of all occupations in the U.S. reported in 2020 [40, 46]. During the COVID-19 pandemic, U.S. nursing homes struggled to maintain sufficient staff and those with staffing challenges had greater difficulty containing the spread of COVID-19 among their residents [47]. Operating under difficult, understaffed conditions put the workforce at risk for depression [48], anxiety [49], and post-traumatic stress disorder (PTSD) [50].

Negative portrayals of nursing homes in the media reported by some participants served to worsen the emotional toll on both staff and nursing home leadership. This is consistent with other studies in the U.S. [25] and Sweden [51]. Curiously, negative media coverage is not new to nursing homes. A study of pre-pandemic newspaper articles about nursing homes in the U. S. from 1999 to 2008 found that most stories were negative or neutral, with only 10.5% positive [52]. Yet during COVID-19, media portrayals were identified by some nursing home leaders as important factors leading to negative emotional well-being.

## Personal experiences of nursing home leadership and staff

Emotional reactions were revealed in this study, mostly in response to the third open-ended question, "How did your professional experiences with the COVID-19 pandemic affect you? What event(s) in your workplace deeply affected you? Please describe the event(s) and the impact on you personally and professionally." Nursing home staff reactions to the COVID-19 pandemic described elsewhere have included fear of infection, burnout, the emotional toll of caring for residents faced with isolation, potential illness, and death, and the impact of critical news reports [25]. Respondents in the present study mentioned all of these reactions as well as emotional stress due to disrupted family relations, disrupted careers, and missed or deferred life milestones. Despite this, participating nursing home leaders described persevering, using scarce resources to stitch together their best solutions while attending to the needs of both their residents and their staff. Abundant self-sacrifice was evident, but this was not without consequence as some respondents experienced personal losses and struggled to maintain passion for their work. Some also experienced significant mental and physical health deterioration. A qualitative Canadian study of nursing home leaders found similar impacts–distress, overwhelming workloads, and emotional consequences including insomnia, exhaustion, anxiety, and depression [53]. A French survey of nursing home staff determined that fear of COVID-19 was related to increased emotional exhaustion among nursing home workers, and increased emotional exhaustion was related to perceptions of decreased quality of care [54]. A unique perspective was also provided by Spanish survey of nursing home workers during COVID-19 that found high levels of job satisfaction despite exposure to resident suffering and death, emotional exhaustion, and job-related pressures. In this study, social support was found to mitigate the negative impact of difficult working conditions [55].

In the current study, it was apparent that many respondents were both traumatized and exhausted as a result of their pandemic experience. In addition to the stresses of caring for residents and staff during the pandemic, many of these respondents expressed exasperation with

chaos introduced by certain government regulations and the punitive nature of enforcement. During this time, nursing homes were often uncertain of how to adapt to U.S. federal and state COVID-19 regulations because regulations were frequently changing, often contradictory, and sometimes lacked clinical sense [25]. Fines could be substantial, straining already tight resources [56]. A noteworthy number of respondents talked of quitting long-term care, and a few expressed regret for having joined the profession at all. Significant staff turnover and burnout were described. Nationally, an estimated 420,000 nursing home employees have left the U. S nursing home workforce since February 2020 [57, 58] and moderate to severe burnout, or emotional exhaustion, was found in over 55% of the nursing home healthcare workers [54]. Emotional stress described by respondents sometimes led to depression. As the COVID-19 pandemic winds down, it is unclear to what extent staff emotional needs are being addressed or what the long-term consequences may be. This raises concerns about maintaining the current and future nursing home workforce in the U.S. and readiness for the next pandemic. New mental health interventions, resources, and supports are needed [59].

## Strengths

This survey reached a national, geographically diverse population and gathered data from a large number of respondents from U.S nursing homes. The qualitative analysis was iterative, carefully conducted, and consensus-oriented to reduce bias.

## Limitations

Since the open-ended questions were part of a self-administered survey, it did not allow for follow-up questions or requests for clarifications. For those who completed the paper survey, the physical space for written responses to open-ended questions was limited. Self-selection of responders would also favor participation by people who had strong opinions or compelling stories to tell.

A 30% response rate for the overall survey was achieved, and although this was not out of line with other recent surveys of nurses (National Nursing Workforce Survey 2020, estimated national response rate 32.6% [60]) our response rate is still less than half the population targeted. Further, among responders, just over half (53.8%) provided answers to the open-ended questions. Response rate was likely affected by the very factors described in many responses to questions about pandemic experiences. The targeted population included nursing home leaders who had just experienced a very stressful and traumatic period of time accompanied by additional reporting requirements due to state and federal COVID-19 regulations. This voluntary survey was yet another demand on their time, and it is reasonable to believe that potential respondents may have been worn down by their clinical and administrative experiences.

## Conclusions

During the COVID-19 pandemic, nursing home residents suffered from significant fears of COVID and death, isolation from family, and loss. New and creative interventions using both old and new technology were successfully implemented to compensate for pandemic-induced isolation. Hallway activities, outdoor activities, and window visits (where possible) were considered successful. Increased social contact with staff was seen as essential, particularly when in-person family contact was prohibited. This placed a greater burden on already overburdened staff and leadership. Ever changing, sometimes punitive government regulations, in many cases, inserted chaos into an already chaotic situation. Leadership and staff put forth extraordinary efforts to address the emergency, and many suffered social, emotional, and physical consequences including COVID-19 exposure. Several respondents planned to quit

their careers. As one respondent stated, "*The pandemic has been a heartbreaking experience, and we will be trying to recover as an industry for years to come*" *(Participant 1020)*. Despite these serious difficulties, some respondents reported they had learned new skills, formed stronger bonds with their staff, and discovered new strategies to address loneliness among their residents that they planned to continue into the future.

## Implications

Upgrading nursing home infrastructure is needed for continuation of new communication technologies including availability of equipment and improved Wi-Fi. Staffing models prioritizing resident contact time and social activities need to be investigated. Punitive approaches to federal, state, and local government oversight need rethinking with an eye toward incentivizing performance improvement. Continued stress as well as post-traumatic stress among staff must be addressed to heal the workforce and make nursing homes a more attractive career. Further research into effective approaches to workforce development and retention is needed. Additionally, focused in-depth interviews of nursing home leadership and staff aimed at exploring pandemic experiences, long term trauma, and other impacts of the COVID-19 pandemic on the mental health of nursing home residents and direct-care workers would provide additional insights important to improving pandemic responses in the future.

## Supporting information

**S1 Appendix. Research protocol.**
(DOCX)

## Acknowledgments

Many thanks to the nursing home directors of nursing and administrators who voluntarily completed this survey and provided their thoughts, perspectives, and powerful stories.

## Author Contributions

**Conceptualization:** Catherine E. Dubé, J. Lee Hargraves, Bill Jesdale, Kate L. Lapane.

**Data curation:** Catherine E. Dubé, J. Lee Hargraves, Carol Cosenza, Kate L. Lapane.

**Formal analysis:** Catherine E. Dubé, Natalia Nielsen, Emily McPhillips.

**Funding acquisition:** Kate L. Lapane.

**Investigation:** Catherine E. Dubé.

**Methodology:** Catherine E. Dubé, J. Lee Hargraves.

**Project administration:** Catherine E. Dubé, J. Lee Hargraves, Carol Cosenza, Kate L. Lapane.

**Resources:** Kate L. Lapane.

**Supervision:** Catherine E. Dubé, J. Lee Hargraves, Kate L. Lapane.

**Validation:** Catherine E. Dubé, Natalia Nielsen, Emily McPhillips.

**Writing – original draft:** Catherine E. Dubé, Natalia Nielsen, Emily McPhillips.

**Writing – review & editing:** Catherine E. Dubé, Natalia Nielsen, Emily McPhillips, J. Lee Hargraves, Carol Cosenza, Bill Jesdale, Kate L. Lapane.

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
