## [Decision Letter · Decision Letter 0]

22 Mar 2023

PONE-D-22-32109US Nursing Home Leadership Perceptions of COVID-19’s Impact on Residents and Staff : A Qualitative AnalysisPLOS ONE

Dear Dr. Catherine,

Thank you for submitting your manuscript to PLOS ONE. After careful consideration, we feel that it has merit but does not fully meet PLOS ONE’s publication criteria as it currently stands. Therefore, we invite you to submit a revised version of the manuscript that addresses the points raised during the review process.

We look forward to receiving your revised manuscript.

Kind regards,

Sally Mohammed Farghaly

Academic Editor

PLOS ONE

Journal Requirements:

Reviewers' comments:

Reviewer's Responses to Questions

**Comments to the Author**

1. Is the manuscript technically sound, and do the data support the conclusions?

Reviewer #1: Partly

Reviewer #2: Yes

2. Has the statistical analysis been performed appropriately and rigorously? 

Reviewer #1: N/A

Reviewer #2: Yes

3. Have the authors made all data underlying the findings in their manuscript fully available?

Reviewer #1: No

Reviewer #2: No

4. Is the manuscript presented in an intelligible fashion and written in standard English?

PLOS ONE does not copyedit accepted manuscripts, so the language in submitting articles must be clear, correct, and unambiguous. Any typographical or grammatical errors should be corrected at revision, so please note any specific errors here.

Reviewer #1: Yes

Reviewer #2: Yes

5. Review Comments to the Author

Reviewer #1: Dear authors,

I am more than pleased to review this manuscript. Thank you for giving such great opportunity.

In general, the topic is interesting and adds a significant contribution to leadership and geriatrics sciences, however, the manuscript need major revisions, organized, and summarized.

Please follow the following comments;

Abstract:

- Research design should be clearly stated

- Number of nursing homes were data are collected must be mentioned

- in setting and participants section : add home to directors of nursing

- Results must be summarized

- No need to add quotes from participants' entries in abstract

Introduction:

- Background is very weak, literature review need to be placed in the international context of the findings in relation to the chosen research problem

- It is better to avoid pronouns referring to authors (replace "we" in the end of introduction with "the current study").

Method:

This section include many details not needed and many important items are missing

- Qualitative design is not clear

- Reasons for choosing the qualitative design to address research problem of this study are not stated - I think if it was conducted as mixed study it will give powerful quantitative impressions about effects on residents and staff, challenges, strategies to mitigate loneliness and social isolation, and benefits.

- Setting should be described enough ( I cannot find simple information as the total numbers of nursing homes where stratums are drawn)

- Much emphasis must be given to participants ( total population; characteristics of participants in terms of job titles, roles in nursing homes; and recruitment measures)

- Causes of low response rate (30%) must be discussed

- Reasons for excluding small nursing homes must be given

- What are the measures used to ensure validity and trustworthy of data collected from subjects?

- Roles of researchers in qualitative analysis process are not needed in method section.

- It is better to avoid pronouns referring to authors

Results:

- Too much details however significant results are not delineated

- It is better to give support to analysis using more quotes from participants self-reporting

- Quantitative analysis if added in results it give significant value especially response s are perceptions

Discussion:

- It does not follow guidelines of effective debate (results of similar and contradictory studies must be given)

- It is better to avoid pronouns referring to authors

- I feel that discussion does not add new than just results

Conclusion:

- Is prefect however, it need to be focused and summarized

Implications:

- Are there no future research directions for this study? I think that perceptions of residents and staff will greatly add value than leaders

Reviewer #2: Reviewer report:

Thank you for the opportunity to review the manuscript. The authors are suggested to apply the following comments to the manuscript:

INTRODUCTION

- Further support of published literature on the research topic nationally and internationally is required.

- A justification of why you used the survey as a data collection method is ultimately needed.

MATERIALS AND METHODS

Survey sample:

- Please explain how the participants were selected.

- Non-participation: Specify how many participants dropped out or refused to participate. Give the reason if applicable.

The survey content:

- Explain more details on the first section as how it was developed, the total number of questions, and the number of questions for each topic

- Describe how the survey was pretested before full implementation

- Lines 124:128 contain unnecessary details

Data collection/ survey process:

Explain the survey process separately after the materials.

Data analysis

- It is better to include a thematic map to justify how the themes and the sub-themes are related.

RESULTS:

- Tables: the illustrated data does not support the aim of the study. It's better to present the demographics of the sample only.

- Results need a major review as most finding need to be supported with participant quotations, in addition to reporting participant number.

DISCUSSION:

- Include the limitation of the study at the end of the discussion.

REFERENCES:

- No 13: Is it unpublished work? It is unclear.

- No 15: rewrite it "Lumivero - Software Solutions for Data Analysis & Management [Internet]. [cited 2022 oct 12]. Available from: " ext-link-type="uri" xlink:type="simple">https://lumivero.com/"

6. PLOS authors have the option to publish the peer review history of their article (what does this mean?). If published, this will include your full peer review and any attached files.

Reviewer #1: No

Reviewer #2: No

---

## [Author Response · Author response to Decision Letter 0]

31 May 2023

Rebuttal – Response to Reviewers

Responses are in BOLD.

Responses to Editor Comments

1. PLOS ONE's style requirements – We have attempted to implement all style requirements as indicated.

2. ETHICS -- Please provide additional details regarding participant consent. In the ethics statement in the Methods and online submission information, please ensure that you have specified what type you obtained (for instance, written or verbal, and if verbal, how it was documented and witnessed). If your study included minors, state whether you obtained consent from parents or guardians. If the need for consent was waived by the ethics committee, please include this information.

Thank you for your careful consideration of subject consent. In this case, our Institutional Review board determined that the study posed minimal risk and was EXEMPT. Their determination was:

 Human research that is exempt under the following guideline(s): 

• 45 CFR 46.104(d)(2) Educational tests/survey/interview procedures, or observation of public behavior. 

With exempt research, formal consent is not required. However, at the beginning of the survey we included the following statements: 

• Please fill out this survey and return it in the postage-paid envelope. 

• Your participation in this study is voluntary. If you come across a question you would rather not answer, feel free to skip it and go on to the next question. 

• Your answers are confidential. No information will be presented or published in any way that would permit identification of any individual or facility. Your name and answers will not be shared with anyone other than the researchers.

• Thank you for taking the time to help with this very important project.

In the preamble to the open-ended questions used for this qualitative study, we also stated the following:

If you would like to tell us more about your COVID-19 experiences at your nursing home, please answer the following questions. If not, please return the survey in the enclosed envelope.

Please reflect upon how you, your facility, and direct care staff responded to resident loneliness and social isolation during the pandemic. We want to hear your candid thoughts and opinions. Your responses will not be linked to your name, but may be reported verbatim along with comments provided by other participants.

We consider voluntary completion of the survey to be evidence of consent and participants who did not consent simply did not fill out the survey. No further written or verbal documentation of consent was obtained or required by the UMass Chan or UMass Boston IRB. We have provided additional clarification in the Human Subjects section.

We have added and clarified the Ethics Statement in the submission portal.

3. Data Availability Important: If there are ethical or legal restrictions to sharing your data publicly, please explain these restrictions in detail. Please see our guidelines for more information on what we consider unacceptable restrictions to publicly sharing data: http://journals.plos.org/plosone/s/data-availability#loc-unacceptable-data-access-restrictions. Note that it is not acceptable for the authors to be the sole named individuals responsible for ensuring data access.

Human research participant data and other sensitive data

For studies involving human research participant data or other sensitive data, we encourage authors to share de-identified or anonymized data. However, when data cannot be publicly shared, we allow authors to make their data sets available upon request.

If there are ethical or legal restrictions on sharing a sensitive data set, authors should provide the following information within their Data Availability Statement upon submission:

Explain the restrictions in detail (e.g., data contain potentially identifying or sensitive patient information)

These qualitative data include highly personal information that could allow identification of participants. In order to obtain the most honest and unbiased responses, we promised responders the following:

• Your answers are confidential. No information will be presented or published in any way that would permit identification of any individual or facility. Your name and answers will not be shared with anyone other than the researchers.

It would be unethical to share our data when we assured our study participants that their data would confidential and that their answers would not be shared. Responses to open-ended questions can be quoted in our manuscript however, due to the statement printed in the preamble to the open-ended questions on our survey:

Your responses will not be linked to your name, but may be reported verbatim along with comments provided by other participants.

Data collection for this research was conducted in 2022 prior to US National Institute of Health requirements for data sharing had taken place. (https://sharing.nih.gov/data-management-and-sharing-policy/about-data-management-and-sharing-policies/data-management-and-sharing-policy-overview)

Provide contact information for a data access committee, ethics committee, or other institutional body to which data requests may be sent

If data is requested for reanalysis or confirmation of results, the petitioner would need to contact our ethics committee: 

Center for Clinical and Translational Science

UMass Chan Institutional Review Board (IRB)

362 Plantation Street,

Ambulatory Care Center, 7th Floor

Worcester, MA 01605-0002

Phone: (508) 856-4261

IRB@umassmed.edu

Response to Reviewers' comments:

Reviewer #1: Dear authors,

I am more than pleased to review this manuscript. Thank you for giving such great opportunity.

In general, the topic is interesting and adds a significant contribution to leadership and geriatrics sciences, however, the manuscript need major revisions, organized, and summarized.

Please follow the following comments;

Thank you for your positive comments and your thorough review. We hope that our clarifications and responses satisfactorily address your concerns.

Abstract:

- Research design should be clearly stated – We provided additional details about the survey design.

- Number of nursing homes were data are collected must be mentioned – Each respondent was from a different nursing home. We clarified – the number of respondents is the same as the number of nursing homes.

- in setting and participants section : add home to directors of nursing – Respondents are now called “nursing home directors of nursing” – these are the leaders of the clinical staff in US nursing homes and are referred to as “Directors of Nursing” or DON.

- Results must be summarized – Thank you. We reviewed this section and believe our results are summarized in the abstract..

- No need to add quotes from participants' entries in abstract -- The quote was removed.

Introduction:

- Background is very weak, literature review need to be placed in the international context of the findings in relation to the chosen research problem – We have added international context and revised the introduction.

- It is better to avoid pronouns referring to authors (replace "we" in the end of introduction with "the current study"). – We have removed the word “we” from the manuscript.

Method:

This section include many details not needed and many important items are missing

- Qualitative design is not clear

- Reasons for choosing the qualitative design to address research problem of this study are not stated – 

The Methods section was revised. Importantly, the following information about the qualitative design was added:

Qualitative inquiry guided by interpretivist epistemology was employed to elicit the experiences of nursing home directors of nursing/administrators during the COVID-19 pandemic. This approach was selected to better understand participant perspectives through stories of personal, staff, and resident pandemic experiences. Stories about strategies employed to address social isolation during COVID-19 were also elicited. Textual responses to optional open-ended survey questions were employed to enhance data collection from a diverse national (U.S.) sample and to maximize personal expression, allowing participants wide latitude to describe strategies, experiences, and the overall impact of the COVID-19 pandemic in their own words. Open-ended questions were appended to a national (U.S.) nursing home survey and were marked “optional.” Open responses and confidentiality protections were intended to minimize response bias. This approach was selected for efficiency and expediency at a time when the COVID-19 pandemic was beginning to wind down and pandemic experiences were still fresh. An iterative reflexive thematic approach to analysis was employed.

I think if it was conducted as mixed study it will give powerful quantitative impressions about effects on residents and staff, challenges, strategies to mitigate loneliness and social isolation, and benefits.

Thank you for your suggestion. We chose to structure this as 2 studies from the outset (as opposed to a single mixed methods study) so that we would have ample space to present the results of each study in separate manuscripts. We intend to provide references to each of these studies within these manuscripts when citation information is available. An initial citation is provided. (Lapane KL, Lim E, DS Mack, Hargraves JL, Cosenza C, Dubé CE. Rising to the occasion: a national nursing home study documenting attempts to address social isolation during the COVID-19 pandemic. JAMDA. Forthcoming.)

- Setting should be described enough ( I cannot find simple information as the total numbers of nursing homes where stratums are drawn)

Thank you. We have included the total number of nursing homes. Additional information about the sample is provided in the section “Survey sample and process”.

- Much emphasis must be given to participants ( total population; characteristics of participants in terms of job titles, roles in nursing homes; and recruitment measures)

Participant characteristics including job titles are provided in Table 1. Characteristics of Participants. There was a small number (4%) of job titles that were neither Director of Nursing nor Nursing Home Administrator. Below Table 1 we have provided additional detail on “Other” job titles. Table 2 provides distribution of participating nursing homes by state. Additional information about the sample is provided in the section “Survey sample and process”.

- Causes of low response rate (30%) must be discussed

The National Nursing Workforce Survey 2020, estimated the US response rate among licensed nurses to be 32.6% . Our response rate is similar. [Smiley et al Journal of Nursing Regulation, April 2021, Volume 12, Issue 1 Supplement https://www.journalofnursingregulation.com/article/S2155-8256(21)00027-2/fulltext ]

We do acknowledge that our response rate may have been affected by the very factors described in many responses to our questions about pandemic experiences. The population we were targeting included nursing home leaders who had just experienced a very stressful and traumatic period of time, accompanied by additional reporting requirements due to state and federal COVID-19 regulations. This voluntary survey was yet another demand on their time, and it is feasible to believe that potential respondents were simply worn down from their clinical and administrative experiences and chose to focus their efforts elsewhere.

We have summarized the above information in the limitations section of the manuscript.

- Reasons for excluding small nursing homes must be given

Small nursing homes (30 beds) were excluded because they differ significantly from larger nursing homes in terms of operations and resources. The average size of a US nursing home in 2022 was 106 beds. [https://www.kff.org/other/state-indicator/average-number-of-certified-nursing-facility-beds/?activeTab=mapcurrentTimeframe=0selectedDistributions=average-number-of-certified-nursing-facility-bedsselectedRows=%7B%22wrapups%22:%7B%22united-states%22:%7B%7D%7D%7DsortModel=%7B%22colId%22:%22Location%22,%22sort%22:%22asc%22%7D] – We have added this explanation.

- What are the measures used to ensure validity and trustworthy of data collected from subjects?

Nursing homes included in our sample were legitimate, Medicare evaluated facilities. Surveys were sent directly to nursing home leadership either by using their official email address or the address of the nursing home. Respondents were all professionals. We were seeking voluntary responses to open-ended questions at the end of the survey for this qualitative study, and thus qualitative respondents were self-selected from the larger group. We have provided a table of characteristics (Table 1) for comparison of open-ended question responders versus non-responders. As noted under Participant Characteristics, “Respondents to open-ended questions tended to be older, have a longer tenure at their current nursing home, and have more time in a leadership position than non-responders.”

**All of the above information already appears in the manuscript.

- Roles of researchers in qualitative analysis process are not needed in method section. – We have removed these roles. 

- It is better to avoid pronouns referring to authors – We have removed pronouns.

Results:

- Too much details however significant results are not delineated

We used a reflexive thematic qualitative analytic approach, qualitative results are descriptive, and there are no tests of significance. To highlight key results we have added a summary table (Table 4). To streamline the results section, we collected most supporting detailed quotes in Table 5.

- It is better to give support to analysis using more quotes from participants self-reporting

We have added a table of themes and illustrative quotes (Table 5).

- Quantitative analysis if added in results it give significant value especially response s are perceptions

Thank you. We agree with your statement – however as a qualitative study we focus on perceptions, opinions and personal experiences.

Discussion:

- It does not follow guidelines of effective debate (results of similar and contradictory studies must be given) 

 Thank you for this consideration. We were following PLOS guidance on Discussion sections (https://plos.org/resource/how-to-write-conclusions/#:~:text=It%20should%20include%3A,your%20results%20and%20initial%20hypothesis. ) which defines the discussion section as “An effective discussion informs readers what can be learned from your experiment and provides context for the results.”

1. the results of your research,

2. a discussion of related research, and

3. a comparison between your results and initial hypothesis. 

We have expanded “compare and contrast to previous studies”. 

- It is better to avoid pronouns referring to authors – We have removed pronouns.

- I feel that discussion does not add new than just results – Please see #1 in PLOS ONE guidance above. We have expanded the “compare and contrast to previous studies” aspect of the discussion section.

Conclusion:

- Is prefect however, it need to be focused and summarized – 

“Is prefect” – Do you mean perfect? If yes, thank you. We have revised the conclusion to be more focused.

Implications:

- Are there no future research directions for this study? I think that perceptions of residents and staff will greatly add value than leaders – 

Thank you for your comment. We believe that further research into effective approaches to workforce development and retention is needed and have added this to the implications section. Other suggestions for additional research are noted in the Discussion section.

Reviewer #2: Reviewer report:

Thank you for the opportunity to review the manuscript. The authors are suggested to apply the following comments to the manuscript:

INTRODUCTION

- Further support of published literature on the research topic nationally and internationally is required.

Thank you. We have revised the introduction and updated the literature cited to provide additional international context.

- A justification of why you used the survey as a data collection method is ultimately needed.

Open-ended questions were added to collect stories and experiences from a diverse sample of nursing home directors of nursing or administrators from across the U.S.. This approach was chosen for efficiency and expediency as the COVID-19 pandemic was beginning to wind down and pandemic experiences were still fresh. 

We have added this explanation to the Research Design section.

MATERIALS AND METHODS

Survey sample:

- Please explain how the participants were selected.

We had included the following information about participant selection in this section (Survey Sample and Process) but have expanded it. Please note that all survey participants were offered the opportunity to respond to open-ended questions: 

First, we acquired a list of all US nursing homes rated by the Centers for Medicare Medicaid Services (CMS) as of August 2021 was acquired. Small nursing homes with 30 beds were excluded. Nursing homes were stratified by size and CMS quality rating into six strata (30-99 or 100+ beds; 1, 2-4, or 5-star ratings) with 283 nursing homes randomly selected in each stratum. Directors of nursing for each nursing home were identified and contact information was collected. Links to the online survey were sent to those with email addresses. Paper surveys including a $5 incentive were mailed to all email non-responders and those without email addresses. An additional $40 was sent after survey completion.

- Non-participation: Specify how many participants dropped out or refused to participate. Give the reason if applicable.

This information was provided in the RESULTS section: “From an eligible sample of 1,676 nursing homes, 504 completed surveys were collected (response rate 30%) with 271 (54% of the 504 respondents) answering at least one open-ended question.” Responding to the survey was voluntary, and responses to open-ended questions were optional. We did not collect information from survey non-responders or reasons why those who did respond did not proceed to answer open-ended questions. Additional information about the characteristics of survey responders who did answer open-ended questions versus those who did not are provided in Table 1.

The survey content:

- Explain more details on the first section as how it was developed, the total number of questions, and the number of questions for each topic

Thank you. We included the total number of quantitative questions in the survey and additional information about topics covered. Further detail about survey questions are reported elsewhere. (Lapane KL, Lim E, DS Mack, Hargraves JL, Cosenza C, Dubé CE. Rising to the occasion: a national nursing home study documenting attempts to address social isolation during the COVID-19 pandemic. JAMDA. Forthcoming.)

- Describe how the survey was pretested before full implementation

Thank you. We have provided a summary of our pretesting process used to revise/finalize the survey.

- Lines 124:128 contain unnecessary details – Thank you for your comment. We believe that transparency regarding limitations to responses is important. As a compromise, we shortened 2 sentences and deleted the others.

Data collection/ survey process:

Explain the survey process separately after the materials.

We have added a heading for the data collection section under “Survey content.”

Data analysis

- It is better to include a thematic map to justify how the themes and the sub-themes are related.

Thank you. We have provided a table with themes, subthemes and illustrative quotes to show these relationships. (Table 5)

RESULTS:

- Tables: the illustrated data does not support the aim of the study. It's better to present the demographics of the sample only.

We respectfully disagree with this comment. We believe that information about how those who responded to open-ended questions are the same or are different from those who did not is important and thus have included both sets of characteristics in Table 1. This information also addresses points raised by Reviewer 1. Other tables provide additional information about the sample, length of open-ended responses, and results. We contend that this information does indeed support the aim of the study.

- Results need a major review as most finding need to be supported with participant quotations, in addition to reporting participant number.

We have added participant numbers to all quotes in the results section and added Table 5 which includes illustrative quotes for each subtheme. Reviewer 1 called for streamlining the Results section – so we opted for a table of longer quotes instead of integrated quotes.

DISCUSSION:

- Include the limitation of the study at the end of the discussion.

Strengths and limitations were included at the end of the discussion section. (Second level heading.) We have expanded this section.

REFERENCES:

- No 13: Is it unpublished work? It is unclear. – This manuscript is currently forthcoming. We have updated the reference. (Lapane KL, Lim E, DS Mack, Hargraves JL, Cosenza C, Dubé CE. Rising to the occasion: a national nursing home study documenting attempts to address social isolation during the COVID-19 pandemic. JAMDA. Forthcoming 2023.)

- No 15: rewrite it "Lumivero - Software Solutions for Data Analysis Management [Internet]. [cited 2022 oct 12]. Available from: https://lumivero.com/"

Thank you. We have updated this reference as you suggest.

---

## [Decision Letter · Decision Letter 1]

9 Jul 2023

PONE-D-22-32109R1U.S. nursing home leadership perceptions of COVID-19’s impact on residents and staff: A qualitative analysisPLOS ONE

Dear Dr. Catherine E Dube,

Thank you for submitting your manuscript to PLOS ONE. After careful consideration, we feel that it has merit but does not fully meet PLOS ONE’s publication criteria as it currently stands. Therefore, we invite you to submit a revised version of the manuscript that addresses the points raised during the review process.

If applicable, we recommend that you deposit your laboratory protocols in protocols.io to enhance the reproducibility of your results. Protocols.io assigns your protocol its own identifier (DOI) so that it can be cited independently in the future. For instructions see: https://journals.plos.org/plosone/s/submission-guidelines#loc-laboratory-protocols. Additionally, PLOS ONE offers an option for publishing peer-reviewed Lab Protocol articles, which describe protocols hosted on protocols.io. Read more information on sharing protocols at https://plos.org/protocols?utm_medium=editorial-emailutm_source=authorlettersutm_campaign=protocols.

We look forward to receiving your revised manuscript.

Kind regards,

Sally Mohammed Farghaly

Academic Editor

PLOS ONE

Reviewers' comments:

Reviewer's Responses to Questions

**Comments to the Author**

1. If the authors have adequately addressed your comments raised in a previous round of review and you feel that this manuscript is now acceptable for publication, you may indicate that here to bypass the “Comments to the Author” section, enter your conflict of interest statement in the “Confidential to Editor” section, and submit your "Accept" recommendation.

Reviewer #1: All comments have been addressed

Reviewer #2: All comments have been addressed

Reviewer #3: (No Response)

2. Is the manuscript technically sound, and do the data support the conclusions?

Reviewer #1: Yes

Reviewer #2: Yes

Reviewer #3: (No Response)

3. Has the statistical analysis been performed appropriately and rigorously? 

Reviewer #1: Yes

Reviewer #2: (No Response)

Reviewer #3: (No Response)

4. Have the authors made all data underlying the findings in their manuscript fully available?

Reviewer #1: No

Reviewer #2: Yes

Reviewer #3: (No Response)

5. Is the manuscript presented in an intelligible fashion and written in standard English?

Reviewer #1: Yes

Reviewer #2: Yes

Reviewer #3: (No Response)

6. Review Comments to the Author

Reviewer #1: Dear authors

Thanks a lot for this great effort appeared in this manuscript

Thanks again for your responses. These responses reflect professional researchers able to upgrade knowledge and practice.

Manuscript is already modified than before; however, literature review needs to be strengthed in terms of research gap, magnitude of the problem, and context of the study. Also, discussion needs to be strengthed.

Reviewer #2: Title: U.S. nursing home leadership perceptions of COVID-19’s impact on residents and staff: A qualitative analysis

Version: 2

Date: 19 Jul 2023

Reviewer's report: The revision has been made appropriately. I have no more issues in this manuscript.

Reviewer #3: (No Response)

7. PLOS authors have the option to publish the peer review history of their article (what does this mean?). If published, this will include your full peer review and any attached files.

Reviewer #1: No

Reviewer #2: No

Reviewer #3: No

---

## [Author Response · Author response to Decision Letter 1]

24 Jul 2023

Response to Reviewers

Paper Title: US Nursing Home Leadership Expereinces with COVID-19 and Its Impact on Residents and Staff: A Qualitative Analysis

Journal Manuscript ID: PONE-D-22-32109 

4. Have the authors made all data underlying the findings in their manuscript fully available?

Reviewer #1: No

Reviewer #2: Yes

Reviewer #3: (No Response)

RESPONSE: It would be unethical to make our qualitative data available as we do not have IRB approval or consent from participants to do so. We have justified this multiple times. If our previously provided detailed justification is insufficient, we will withdraw and submit to another journal.

1. If the authors have adequately addressed your comments raised in a previous round of review and you feel that this manuscript is now acceptable for publication, you may indicate that here to bypass the “Comments to the Author” section, enter your conflict of interest statement in the “Confidential to Editor” section, and submit your "Accept" recommendation.

Reviewer #1: All comments have been addressed

Reviewer #2: All comments have been addressed

Reviewer #3: (No Response)

RESPONSE: Reviewer 1 indicates that all comments in the previous review round have been addressed. However, new requests for revisions (particularly for the Introduction and Discussion) are included below. Most of the previous review comments focused on these sections as well. We believe we have addressed the requested revisions and making further demands for additional revisions is unwarranted. We have, however, made a good faith effort to address Reviewer 1’s issues.

Reviewer #1: Dear authors

Thanks a lot for this great effort appeared in this manuscript

Thanks again for your responses. These responses reflect professional researchers able to upgrade knowledge and practice.

Manuscript is already modified than before; however, literature review needs to be strengthed in terms of research gap, magnitude of the problem, and context of the study. Also, discussion needs to be strengthed.

REVIEW 

This study wants to examine experiences of nursing home leadership addressing resident loneliness and social isolation during the COVID-19 pandemic in the US and personal and professional impacts on themselves and staff.

I think that the focus of this study could be important. Congratulations for your work and effort. 

Thank you. However, we are concerned about the lengthy time required for PLOS ONE reviews and revisions resulting in delay of publication in PLOS ONE or elsewhere and ultimately reducing the relevance and impact of our work.

However, I would like you to pay attention to the following indications for improvement its importance.  

Abstract: 

- Why does the abstract include the number of the participant who has said the sentence(“The pandemic has been a heartbreaking experience, and we will be trying to recover as an industry for years to come”), and the text, in the results section, does not do the same with the rest of the participants and their sentences?

RESPONSE: Participant numbers do appear for all quotes in the Results section. It is unclear what this comment is referring to.

Introduction: 

Based on the information presented in the introduction, key information missing pertains to the following: 

- The introduction is very weak. I need the following questions (that make up the introduction) to be answered in more depth: What is the problem? Why is it important? What is known so far (scientific articles)? What is not known? Link to the objectives of the study. 

RESPONSE: Much of this information already appears in the Introduction. We affirm that we have stylistic differences with Reviewer 1 but are making a good faith effort to address Reviewer 1’s issues. What already appears in the introduction is printed in bulleted list below. New revisions are INDENTED.

What is the problem? 

• Nursing homes were a vector for COVID-19

• 1.3. million nursing home residents were affected

 • Loneliness and social isolation have been associated with negative health outcomes in general and among older people in congregate care settings such as nursing homes

• Loneliness and isolation among nursing home residents were exacerbated by COVID-19

• Staff were severely affected both by both illness and working conditions

 • Nursing homes lost staff at high levels and staff shortages continue to be a problem

Why is it important? 

• National studies in the US addressing the actions taken by nursing home leadership to address social isolation and the personal and professional impact of the pandemic on nursing home leaders, residents, and their staff are lacking

 • Our study intends to further the collective understanding of these issues that may be pertinent to future pandemic preparedness

What is known so far (scientific articles)? 

Integrated within the Introduction:

• Nursing home leaders rapidly developed new solutions to deal with a drastically worsening situation

• evidence-based solutions were not available

• working conditions were difficult in nursing homes and there was substantial psychological impact on the workforce

 • Loneliness and social isolation have been associated with negative health outcomes including all cause mortality

What is not known? 

• only a few national studies in the U.S have documented how U.S. nursing homes pivoted to provide care

• only a few national studies in the U.S. have documented how leadership, staff, and residents were impacted by the COVID-19 pandemic

Link to the objectives of the study. 

• Aims were to explore strategies employed to address resident loneliness and social isolation in U.S. nursing homes and to describe the personal and professional impact of the pandemic on nursing home leaders, facilities, and their staff from the perspective of directors of nursing or administrators.

Methodology 

Information provided in the methods and procedures is not detailed or clear enough to replicate the study. 

RESPONSE: We respectfully disagree. We provide a lot of detail, some of which was removed in response to Reviewer 1 comments in the last review. ( “Roles of researchers in qualitative analysis process are not needed in method section.” [Reviewer 1, Initial Review])

In particular, we describe:

1 – Use of a national survey to obtain responses to 3 open-ended questions designed to elicit stories about strategies employed to address social isolation among nursing home residents during COVID-19 and how professional experiences with the COVID-19 pandemic affected the respondent.

2 – Participant selection/recruitment

3 – Data collection approach

4 – The wording of the open-ended questions (we have provided additional clarification)

5 – A general description of the other survey questions (reference to survey manuscript provided)

6 – Survey question testing procedure

7 – Data collection approach

8 – Qualitative analysis approach including: use of modified inductive thematic analysis; an iterative process of emersion/crystallization; preparation of a consensus summaries; preparation of theme lists; detailed coding of cross-cutting themes.

We did not intend to teach readers how to conduct qualitative research. Our assumption is that if someone had qualitative research skills, they could replicate our work. If they were lacking qualitative research skills, they would need to seek out training to obtain those skills prior to replicating our work.

** However – in the spirit of a good faith effort, we have added a RESEARCH PROTOCOL as supplementary material.

See below for suggestions/ feedback. Also, we strongly recommend you to use the Consolidated Criteria for Reporting Qualitative Research guidelines and

The Working Group on Journal Article Reporting Standards for Qualitative Research.

We are very familiar with the COREQ checklist and used it in the conduct and reporting of this research. The APA article you refer to above is a good reference and helpful in preparing a manuscript for a qualitative study. As a group, we have published multiple qualitative articles over the years. The details of the APA guidance attempts to address all qualitative research, but there are many different kinds. In this exploratory qualitative research study, we are not generating theory. Rather, we are reporting on the experiences of a population of people who lived through an unprecedented crisis unique in modern times.

- Please, begin this section with Theoretical framework

RESPONSE: We have added a conceptual model for the phenomenon under study based on Rapaccinia, et. al. 2020. Although this was developed for industry, the stages mirror what we anticipated the nursing home experience to be. (Rapaccini M, Saccani N, Kowalkowskic C, Paiolae M, Adrodegari F. Navigating disruptive crises through service-led growth: The impact of COVID-19 on Italian manufacturing firms. Industrial Marketing Management 2020; 88: 225–237.)

- What methodological orientation was stated to underpin the study? e.g. grounded theory, discourse analysis, phenomenology, content analysis.

RESPONSE: As stated in our manuscript “This exploratory qualitative inquiry [is] guided by interpretivist epistemology” and “A modified inductive thematic analysis approach, guided by principles of experiential reflexive thematic analysis, was employed.” We did not employ grounded theory, discourse analysis, phenomenology, or content analysis. We employed INTERPRETIVIST EPISTEMOLOGY and THEMATIC ANALYSIS. 

Interpretivist Epistemology 

Kivunja C, Kuyini AB. Understanding and applying paradigms in educational contexts. International Journal of Higher Education. 2017; 6(5):26-41.

Thematic Analysis:

Braun V, Clarke V. Thematic Analysis: A Practical Guide. London: Sage Publications Ldt, 2022. 

Guest G, MacQueen KM, Namey EE. Applied Thematic Analysis. Thousand Oaks CA: Sage Publications Inc., 2012.

Tolley EE, Ulin PR, Mack N, Robinson ET, Succop SM. Qualitative Methods in Public Health: A Field Guide for Applied Research, Second Edition. San Francisco CA: John Wiley Sons, 2016. 

- Participant selection: in this section clarify the following issues and not in another section. 

We have renamed the “Survey Sample and Process” section “Participant Selection and Survey Process”

o How were participants selected? e.g. purposive, convenience, consecutive, snowball.

RESPONSE: We have described this. We have added clarification that the national sample was purposive while the final qualitative sample was a convenience sample comprised of those from the national sample who voluntarily responded to open-ended questions. This convenience sample still had good national representation (see Table 2).

o How many participants were in the study? Sample size calculation.

REPSONSE: There is no sample size calculation in qualitative research. Adequate sample size is determined by SATURATION. Depending on the study and the data collected, saturation may be achieved with as few as 15 participants or fewer. We address saturation at the beginning of the Results section – which is where that discussion belongs. 

Although we consider the number of participants to be part of our Results, we have added the number (271) to this Participant Selection section as well. It also appears in Table 1.

o How many people refused to participate or dropped out? Reasons?

RESPONSE: We have indicated that 271 of 504 survey responders completed the optional open-ended questions. We have no way of knowing why some people chose to skip these optional questions. No one dropped out.

- Setting: use this section to talk about Where was the data collected? e.g. home, clinic, work place; 

RESPONSE: We indicated that surveys were sent to work email or postal addresses. However, participants could have completed these surveys anywhere. 

Was anyone else present besides the participants and researchers?; 

The researchers were not in the room as survey responses were collected online and by US postal service. It is unknown if others were present besides the participant. 

What are the important characteristics of the sample? e.g. demographic data, date.

Please see tables 1 and 2.

- Add headings such as Data analysis and rigor, Data collection

REPSONSE: We already have these sections. The first is called “Qualitative analytic approach” to which we have added: “and Rigor” – the heading Data Collection appeared in the last version of the manuscript.

Results

- Please make the tables (at least table 1) smaller and more dignified by highlighting or bolding important headings, such as age, gender, etc.

RESPONSE: It is unclear what is meant by “more dignified” tables. Important headings, such as age, gender, etc. were already bolded. We indented the cells below each heading to allow the headings to stand out more.

We also combined some categories in table 1 to make it a little shorter.

- Table 2, make it smaller.

RESPONSE: This table shows the response distribution by US state. There are 50 US States so we cannot make the table smaller. We added columns to make the table more compact.

- Table 3, idem. Add it the standard derivation.

RESPONSE: This table is already small (3 rows by 5 columns – and one of the columns was something you asked us to add.)

We are assuming that what you meant in the second part of your comment was “Add standard deviations.” We have rechecked our calculations and updated this table. We have included standard deviations.

Discusión

- It would be good starting with the purpose of the study. 

We have added the purpose to the beginning of the discussion section.

- In my opinion, there should be more scientific discussion with more references to scientific articles.

- Discuss the themes as you have placed them in the results.

RESPONSE: We have revised the Discussion section once again, adding reflections on the conceptual model, headings corresponding to the 3 major themes, and additional references to scientific literature. Currently there are 32 citations to published literature in the Discussion section.

References

- Only 22 references is too few.

RESPONSE: There were 51 references provided, not 22. There were 27 references in the Introduction alone. In this most recent revised version there are 61 total references.

- Review the bibliography guidelines according to the journal's standards. Not all of them are well referenced

RESPONSE: It is unclear what this comment means. All citations are in proper format according to the journal’s guidelines and include all bibliographic information derived primarily from PubMed. A PLOS ONE editor has reviewed the reference list, and only indicated concern for one article that contained no page numbers (there are no page numbers for that article – it is an on-line article and the citation was provided by the publisher with no page numbers). 

One reference that was, until now, listed as “forthcoming” has been updated with a complete citation as this article (submitted to another journal after initial submission of this article to PLOS ONE) has been published and is now available online.

On journal guidelines previously referenced only doi number was required, however we have added PMCID and PMC numbers wherever they were available.

---

## [Decision Letter · Decision Letter 2]

11 Oct 2023

U.S. nursing home leadership experiences with COVID-19 and its impact on residents and staff: A qualitative analysis

PONE-D-22-32109R2

Dear Dr. Catherine,

We’re pleased to inform you that your manuscript has been judged scientifically suitable for publication and will be formally accepted for publication once it meets all outstanding technical requirements.

Kind regards,

Sally Mohammed Farghaly

Academic Editor

PLOS ONE

Additional Editor Comments (optional):

Reviewers' comments:

Reviewer's Responses to Questions

**Comments to the Author**

1. If the authors have adequately addressed your comments raised in a previous round of review and you feel that this manuscript is now acceptable for publication, you may indicate that here to bypass the “Comments to the Author” section, enter your conflict of interest statement in the “Confidential to Editor” section, and submit your "Accept" recommendation.

Reviewer #3: All comments have been addressed

2. Is the manuscript technically sound, and do the data support the conclusions?

Reviewer #3: Yes

3. Has the statistical analysis been performed appropriately and rigorously? 

Reviewer #3: Yes

4. Have the authors made all data underlying the findings in their manuscript fully available?

Reviewer #3: Yes

5. Is the manuscript presented in an intelligible fashion and written in standard English?

Reviewer #3: Yes

6. Review Comments to the Author

Reviewer #3: (No Response)

7. PLOS authors have the option to publish the peer review history of their article (what does this mean?). If published, this will include your full peer review and any attached files.

Reviewer #3: No

---

## [Editor Report · Acceptance letter]

31 Oct 2023

PONE-D-22-32109R2 

U.S. nursing home leadership experiences with COVID-19 and its impact on residents and staff: A qualitative analysis 

Dear Dr. Dubé:

I'm pleased to inform you that your manuscript has been deemed suitable for publication in PLOS ONE. Congratulations! Your manuscript is now with our production department. 

Kind regards, 

on behalf of

Professor Sally Mohammed Farghaly 

Academic Editor

PLOS ONE